# Exploring nurses' experiences of a tailored intervention to increase MMR vaccine acceptance in a Somali community in Stockholm, Sweden: a qualitative interview study

Emma Appelqvist [1,2] Asha Jama,[1,3] Asli Kulane,[3] Adam Roth,[1,2] Ann Lindstrand,[4] Karina Godoy-Ramirez[5]

For numbered affiliations see end of article.

**Correspondence to**
Emma Appelqvist;
emma.appelqvist@
folkhalsomyndigheten.se

## ABSTRACT

**Objectives** To explore nurses' experiences of a tailored intervention that supported them with knowledge and tools to use during encounters and dialogue with parents with low vaccine acceptance.

**Design** A qualitative study with in-depth interviews conducted in 2017. Data were analysed using thematic analysis.

**Setting** This study was part of a multicomponent intervention targeting Somali parents and the nurses at child health centres in the Rinkeby and Tensta neighbourhoods of Stockholm. An area with documented low measles, mumps and rubella (MMR) vaccination coverage. Previous research has revealed that Somali parents in the community delayed MMR vaccination due to fear of autism despite lack of scientific evidence. The interventions were implemented in 2015–2017.

**Participants** Eleven nurses employed at the child health centres involved in the intervention participated in interviews. The tailored intervention targeting nurses included a series of seminars, a narrative film and an information card with key messages for distribution to parents.

**Results** The qualitative analysis revealed an overarching theme: perception of improved communication with parents. Two underlying themes were identified: (1) feeling more confident to address parents' MMR vaccine concerns and (2) diverse tools as useful support to dispel myth and reduce language barriers.

**Conclusion** From the nurses' perspective, the tailored intervention was useful to improve communication with parents having vaccine concerns. Nurses have a crucial role in vaccine uptake and acceptance. Interventions aiming to strengthen their communication with parents are therefore essential, especially in areas with lower vaccine acceptance.

## INTRODUCTION

Cases of measles increased in Europe and the WHO European Region in 2017–2019, although the disease could be prevented and even eradicated by effective vaccines.[1 2] In the

---

## STRENGTHS AND LIMITATIONS OF THIS STUDY

⇒ Qualitative interviews and thematic analysis enabled a deeper understanding of nurses' experiences of a tailored intervention supporting encounters and dialogue with parents having low vaccine acceptance.

⇒ In-depth interviews of nurses provided unique insights into their experiences of using tailored tools.

⇒ This study is limited as it only focused on the perspective of the nurses; further studies are needed to capture the perspective of the parents.

⇒ The results are relevant for the specific community and setting for the study; caution should be used in attempting to generalise the findings to other settings.

---

WHO European Region, more than 82 000 cases were reported.[2] The majority of the cases occurred among underimmunised individuals and in pockets of lower vaccination coverage below the recommended threshold of 95% coverage for achieving herd immunity.[1 2]

The WHO Regional Office for Europe developed an approach, the tailoring immunisation programmes (TIP), aiming to facilitate a deeper understanding of barriers and drivers for suboptimal vaccination coverage in particular communities.[3] Based on findings of barriers and drivers, tailored and targeted interventions may be designed and implemented to support improved and equitable vaccination uptake.[4] The TIP approach is based on scientific evidence from social and behavioural science research, underpinned by the Capability-Opportunity-Motivation-Behaviour model and the Behaviour Change Wheel framework, which has been adapted for behavioural insights for vaccinations.[3 5]

In Sweden, the measles, mumps and rubella (MMR) vaccination coverage has been high

and stable throughout the last decade, with 97.1% of 2-year-old children vaccinated against MMR in 2020.[6] The national immunisation programme in Sweden is voluntary, and all vaccinations are offered free of charge. The first dose of MMR vaccination is offered at 18 months at child healthcare centres (CHCs), and a second dose is offered by school health services in grades 1 and 2 (children aged 7–8 years). Cases of measles in Sweden have varied from 3 to 51 cases per year during the last 10 years.[7] All recent cases have either been imported or epidemiologically linked to an imported case, apart from a few sporadic cases of unknown origin (Hélène Englund, personal communication, The Public Health Agency of Sweden, 2022). The regional verification committee of WHO concluded measles and rubella to be eliminated in Sweden in 2015 and 2017, respectively.[8] There are, however, regional and local pockets in Sweden with lower MMR vaccination coverage such as, Rinkeby and Tensta districts in the region of Stockholm where there is a high percentage of residents with foreign backgrounds, approximately 30% of these are of Somali origin.[9] In 2000, the vaccination coverage for MMR for children aged 2 years in Rinkeby and Tensta started to decline, and between 2002 and 2013, the vaccination coverage stabilised at around 70%. With the accumulation of susceptible unimmunised populations, the Rinkeby and Tensta area had an increased risk of measles outbreaks. Between January and March spring 2017, there was a measles outbreak in the area, which afflicted the Somali community. During the outbreak 12 cases of measles were reported, of which nine cases were children.[10] In 2013, the Public Health Agency of Sweden piloted the TIP approach and conducted formative research (TIP phase 1–2, based on the TIP Guide version 1, 2013) in three communities with known or suspected suboptimal MMR vaccination coverage: anthroposophic, Somali and undocumented migrants.[11–14]

Results from previous qualitative studies among parents in the Somali community in this area showed that parents had low vaccine acceptance and delayed MMR vaccination primarily due to fear that children would 'stop talking' and develop autism.[12] Concerns about vaccine-induced autism due to MMR vaccination originate from the spurious, retracted article published by Wakefield *et al*[15] in 1998, which suggested a link between the MMR vaccine and autism. No evidence for such a link between autism and MMR vaccines has been found in several extensive international studies.[16 17] Despite the lack of evidence, the myth has been rooted among Somali parents in Rinkeby and Tensta for years.[12] Similarly, lower vaccination coverage for MMR has been observed in Somali communities in the USA, again due to fear of autism and possible links to the vaccine.[18 19] Previous studies have shown that parents in Rinkeby and Tensta would attend all appointments at the CHC but refrain from attending the 18 months appointment at which the MMR vaccine is offered. Thus, access and convenience for vaccination services were not deemed to be a barrier for parents with low vaccine acceptance as parents attended their CHC. Moreover, it became evident that nurses at the CHC did not address parents' concerns regarding MMR vaccination.[13] Previous research has highlighted the crucial role of nurses in parents' decision making about vaccinations.[20–23] As, having an open and accepting approach is key in discussions regarding vaccinations,[21 22] multicomponent tailored interventions were therefore designed and implemented in 2015–2017 for two main target groups in Rinkeby and Tensta: Somali parents and CHC nurses. The design and implementation of these interventions have been described in detail previously.[24] Although the intervention project had two main target groups, this study focuses only on the evaluation of those intervention activities for nurses. This current study aimed to explore nurses' experiences of a tailored intervention that supported them with knowledge and tools to use during encounters and dialogue with parents with low vaccine acceptance.

## METHODS

### Intervention activities and tools

The intervention activities for nurses included education and training as well as a narrative film with Somali role models and information card to use as tools when interacting with parents. The overall design and implementation of the intervention for both parents and nurses have been described in previously published research.[24] All 12 nurses working at the CHCs in either Rinkeby or Tensta at the time of implementation were offered and accepted to participate in the intervention activities. During three consecutive weeks in August and September 2015, all nurses were invited to a series of seminars. Each seminar lasted 2 hours and covered different topics relating to MMR vaccination, that is, measles, MMR vaccination, communication and autism. The focus was particularly on the epidemiological situation and MMR vaccinations both in the context of Somalia, from where the parents originate, and in the context of Sweden where the parents live currently. In addition, an expert on autism presented the current knowledge of autism and its causes. How to use scientific evidence to debunk the link between autism and MMR vaccination, how to communicate with parents with low vaccine acceptance and culturally tailor communication were also covered in the seminars. At the end of the third and final seminar, nurses attending the seminar (n=*12*) were asked to participate in a short survey with 11 questions for evaluation purposes. The result of the survey was used to inform the interview guide.

The intervention also included a 14 min narrative film with Somali role models and experts, entitled 'Vaccination – a wise choice for your child'.[24] The film included information about measles and MMR vaccination as well as the experiences of parents, a nurse and a religious leader. The film was in Somali with Swedish subtitles. The nurses were given a link to the film when it was published on YouTube in 2016.[25] A two-sided information card was

provided to nurses to use in their consultations and to share with parents. The card consisted of key messages in the Somali language on one side with the same information on the other side of the card in Swedish.[25] The key messages focused on: vaccinations offered free of charge for all children, the importance of childhood vaccination to protect against infectious diseases and remember to vaccinate your child before travelling abroad. In addition, a link and QR code to the narrative film and a website with further information were included on the card.[25]

## Qualitative interviews

### Setting and participants

In the fall of 2017, all 12 nurses employed at the CHC in Rinkeby and Tensta were invited to participate in qualitative in-depth interviews. The senior paediatrician at the Regional Preventive Child Health Services facilitated the invitation and recruitment into the study. Three nurses employed after the seminar series held in 2015 were briefed about the seminars and given the materials (film and information card) used in the intervention. Hence, they were deemed eligible for the study and invited to participate in the interviews. Participants were informed in advance about the purpose, design and methodology of the study. All interviews were conducted in November 2017. In-depth interviews were held with 11 of the nurses who consented. One nurse was on leave during the study period. All nurses participating were female and had a degree in nursing with a specialisation in children and youth, which is a requirement to work at CHCs as a nurse. The nurses' working experience at CHCs generally ranged from 3 months to 19 years, with a median of 6 years. Specifically, the nurses had been working at the CHCs in either Rinkeby or Tensta from 3 months up to 17 years, with a median of 4 years, at the time of data collection. Native background of the nurses varied but was not part of the data collection in order to protect the identity of the nurses in this small sample size.

### Data collection

The interviews were semistructured using a thematic guide (see online supplemental file). The outline was adapted from a previous study and also informed by the survey previously conducted after the final seminar.[13] The thematic guide included questions regarding experiences of the interventions, benefits and changes in daily work since the intervention as well as the perception of parents' experiences of the interventions. The three nurses who were employed at the CHCs after the seminar series had been held were only asked questions about the parts of the tools in the intervention they had been able to use (film and card), and questions about the seminars were excluded. The interview guide was piloted by two members of the research team with minor adaptions made with additional probing questions and reorganising of a few questions for a better flow. Interviews were conducted in Swedish by the two authors (EA and KG-R) and digitally recorded. All interviews were conducted in a private space in the workplace, chosen by the interviewee to allow for open discussions. The interviews ranged from 18 to 43 min, 34 min on average. Written informed consent was obtained before the start of each interview.

### Data analysis

All interviews were transcribed verbatim, and analysis was conducted using a thematic analysis.[26] The thematic analysis allowed the analysis to focus on purposely chosen analytic interests regarding the intervention implemented and the tools provided for the nurses. The first author (EA) read the interview transcripts several times to become familiar with the content and thereafter coded by the first author. Two authors (EA and KG-R) assessed and discussed initial coding for consistency after which themes were developed. Themes and initial coding were shared among the research team to elaborate and revise the themes until a final version was agreed on. The analysis was conducted both manually and in Microsoft Office Excel. An example of the analytic process is provided below (table 1) and suitable quotes were selected to support the findings, which are presented in the results.

### Patient and public involvement

None.

## RESULTS

An overarching theme emerged, perception of improved communication with parents and two underlying subthemes were identified:

► Feeling more confident to address parents' MMR vaccine concerns.
► Diverse tools as useful support to dispel myth and reduce language barriers.
  – Tailored seminars.
  – The information card.
  – The film.

| Table 1 | An example of the analytic process | | |
|---|---|---|---|
| Text | Code | Subtheme | Overarching theme |
| But then it probably has with a lot of things to do, maybe with me, that I'm secure in my professional role and also all the work that has been done in various ways. | Different things that make me secure in my professional role. | Feeling more confident to address parents' MMR vaccine concerns. | Perception of improved communication with parents. |
| MMR, measles, mumps and rubella. | | | |

## Perception of improved communication with parents

### Feeling more confident to address parents' MMR vaccine concerns

The nurses expressed that they felt more confident in engaging in a dialogue with parents discussing MMR vaccination and autism after the intervention. Some nurses described being generally more secure in their professional role, whereas other nurses explicitly described being more confident to engage in discussions due to increased knowledge.

> The more knowledge I have, the more confident I am to engage in discussions. (Nurse, #1)

Nurses described that there had been a change in the way they feel regarding discussing MMR vaccination and autism. After the intervention, they felt that they can say with confidence that there is no scientific evidence for the vaccine to cause autism, whereas before the intervention, some nurses expressed feeling insecure and hesitated to vaccinate the child if the parents had low vaccine acceptance. Before the intervention, some nurses described that, instead of engaging in a dialogue in such a situation, they just offered the possibility to postpone the MMR vaccination. As two nurses described:

> '[W]hat if I vaccinate and they get autism?' I was feeling guilty… now I know, it doesn't cause autism. (Nurse, #9)

> [T]ry to differentiate and say that autism is a separate thing, which I also can explain if you'd like, but the vaccine and the disease are not linked. (Nurse, #8)

With reinforced and increased knowledge, nurses felt more secure and confident in saying that there is no scientific evidence for a link to autism, which in turn enabled a better discussion with parents. The dialogue between nurses and parents was facilitated as nurses now felt they could respond to the concerns of parents having low vaccine acceptance and not only describe the MMR vaccine but also confidently communicate the strong scientific evidence of the vaccine not causing autism. When parents brought up that children could become silent due to the vaccine, nurses could name autism and also had the knowledge to explain autism to the parents, elaborating the dialogue and responses to parents.

> [F]eels safe to directly say that there is no evidence. (Nurse, #4)

Although the nurses described positive changes, there were still challenges in the dialogue with parents. An issue raised was the delicate balance between discussing proactively versus trying to persuade the parents. In challenging discussions with parents, the nurses did not want to feel like they were a salesperson for the vaccine or persuading the parents in their vaccination decision instead of enabling the parents to make a voluntary decision themselves.

> I don't want to feel like a salesman when it's the most challenging. (Nurse, #5)

## Diverse tools as useful support to dispel myth and reduce language barriers

### Tailored seminars

The eight nurses taking part in the seminars were asked questions about their experiences in the interviews. Overall, the nurses expressed that the seminars were beneficial and valuable. The structure and format of the seminars were satisfactory and well received. Moreover, the nurses highlighted that the content and topics included were highly relevant and useful. Particularly, the nurses' remembered the seminar about the epidemiological situation of measles in Somalia, the Somali vaccination programme and how parents in and from Somalia reason regarding vaccinations as insightful. They also remembered the seminar about autism to be helpful. Gaining cultural insights as well as useful phrases to use in dialogue with parents were particular highlights from the seminar series. Moreover, nurses also emphasised that reinforcing that autism is a disorder with a complex aetiology with a hereditary component has helped them to describe autism to parents having low vaccine acceptance and also debunk the myth regarding the MMR vaccine.

Nurses felt they have been able to take the discussions with parents to another, deeper level after the seminars. They felt more knowledgeable and had practical tools for how to discuss MMR vaccinations and measles in a more structured way with the parents. Some highlighted that learning 'one-liners' had been especially useful tools, whereas others felt they had more evidence-based information as arguments.

> I received many tools for [use] when I came to work. (Nurse, #4)

The nurses expressed that they had increased their knowledge and could dispel the myth of a link between MMR vaccination and autism in their dialogue and discussion with parents. They emphasised that it was very valuable to learn more about the scientific evidence and large epidemiological studies that have been conducted regarding MMR vaccinations and autism. Knowing the thorough research and the current solid evidence base made the nurses feel more confident in their own beliefs and in their conversations and in conveying the information to the parents. The nurses found new ways to approach and motivate the parents and were able to handle the discussion differently when parents had low vaccine acceptance and were worried about autism and MMR vaccination. In addition, nurses felt more knowledgeable regarding autism, its early signs and symptoms, and the complexity of factors potentially involved in the onset of the disease, despite the aetiology not yet being clear.

> [H]ad the greatest benefit of being able to separate and explain what autism is and what the vaccine is, in my dialogues with parents. (Nurse, #6)

I have found good arguments based on research and [I] can confidently tell it to others, indeed. (Nurse, #10)

### Information card

Nurses described the usefulness of the card and how it had been helpful to hand it out directly to the parents. They particularly highlighted the link to the film and recommended parents to watch the film at home. Moreover, a stack of cards could lay on the nurse's desk, making it highly visible and easier to remember to hand it out. By handing out the card with information in both Swedish and Somali, the nurses knew exactly the content of the information they gave to the parents directly. In cases of relying on interpreters, some of the nurses were concerned that some of the interpreters may not know how to translate fully the information regarding the diseases and vaccinations correctly and hence the parents may not have received all the information. Thus, by using the card, the nurses ensured the parents received the essential and correct information.

[it has] been really good to hand something over and ask them to watch the film and then come back. (Nurse, #9).

### The film

Nurses perceived the film positively, particularly since it was a visual tool and available in the Somali language, decreasing language barriers and facilitating a better understanding of the content. The nurses had different ways of using the film. Many nurses recommended parents to watch the film at home with their spouse and also other family members, relatives or friends, and then they would discuss any questions or concerns raised during the next appointment at CHC. The information card containing the link to the film was often given as a way to promote the film. Other nurses watched the film together with the parents during consultations. Some nurses thought that the film was too long to watch together with parents during appointments as they have a lot to cover during the appointments and not enough time to show a film of 15 min. At one of the CHCs, the film was shown on repeat in the waiting room so that parents could watch while waiting for their appointment. On occasions, nurses could sit and watch the film together with parents in the waiting room, although some nurses thought it was not an optimal approach as they believed they might single out parents as having low vaccine acceptance.

Nurses emphasised that the film helped parents to reconsider their decision for MMR vaccination and five nurses recalled and gave examples of situations where parents with low vaccine acceptance had changed their mind after watching the movie and chose to vaccinate their child against MMR. Other parents may not have changed their decision regarding MMR vaccination, but the nurses felt that they were less sceptic towards the vaccine.

[C]ame back to vaccinate after having watched the film. (Nurse, #1)

[S]ome become softer, the film is really helpful actually. (Nurse, #4)

According to the nurses, the parents believed the film was a valuable tool because it included relevant evidence-based information and highlighted the perspective of the parents. Moreover, the film was perceived as credible as most of the experts and parents participating in the film were of Somali origin and spoke Somali. The experts were also well known, trusted and respected authorities in the community.

Nurses expressed that they perceived parents to be less questioning and more open to discussions related to MMR vaccination after the interventions. Parents had become more open to receiving information and having a dialogue. Moreover, the nurses also felt that parents had gained more knowledge, in particular about autism and posed fewer questions about the disease.

[it is] difficult to measure, it is just a feeling I have. (Nurse, #1)

[I] believe that there are more who are knowledgeable about autism, actually. (Nurse, #8)

In addition, the card and the film were especially useful tools during the outbreak of measles after the intervention, as expressed by this nurse:

During the outbreak we tried to reach out in every way by using information cards, the film, visit to open day care, pediatricians informing… (Nurse, #7)

## DISCUSSION

Findings from the present study suggest that tailored interventions with culturally tailored materials can facilitate nurses' communication about vaccinations with parents of Somali descent having low vaccine acceptance. From the perspective of the nurses, the tailored intervention using diverse tools supported the dialogue, dispelled myth and reduced language barriers while also giving nurses cultural insights. Importantly, nurses gained additional confidence in the scientific evidence supporting that MMR vaccination does not cause autism, and they were now able to dispel the myth in their dialogue and better convey this information to the parents. The seminars both provided and reinforced knowledge that boosted the nurses' confidence in discussing autism, which made them feel more secure when communicating with the parents. Following the intervention and by using the tailored tools provided, the nurses engaged in discussions regarding MMR vaccinations and autism with parents in a different way. Before the intervention, some nurses hesitated themselves and did not address parents when having concerns regarding MMR vaccinations.[13] In addition, tools were also useful during the outbreak of measles in 2017.

Before the intervention was implemented, key barriers were parents fearing that their children would 'stop talking' following an MMR vaccination, and some parents also described unpleasant encounters with nurses.[12 13] However, a key facilitator for vaccinating parents in the community was their trust in nurses.[12 13] Autism was a sensitive topic that both parents and nurses sidestepped when discussing MMR vaccinations. Similar barriers to vaccination due to fear of autism has also been observed in a Somali community in the USA.[18 19]

Factors influencing suboptimal vaccination coverage are context specific, complex and multidimensional.[27] Factors affecting declining vaccine acceptance include barriers in availability and convenience in healthcare systems as well as social determinants, social and cultural norms.[28] In addition, these factors are often specific for their context, setting and type of vaccine, and they may also change over time.[27] Limited access to healthcare and vaccination services for populations has also influenced suboptimal vaccination coverage in Europe.[29]

Using the TIP framework for this intervention project was useful for gaining a thorough understanding of the barriers and drivers for MMR vaccination in the Somali community.[24] Previous studies have highlighted the crucial role of nurses and healthcare providers in parental vaccine acceptance, trust and decision making.[30–33] The belief and attitudes of healthcare workers are essential to parental vaccine acceptance and the demand for childhood vaccinations.[34 35] Hence, it was deemed necessary to tailor interventions for the nurses specifically as part of the multicomponent intervention project and to complement the interventions targeting the parents. Multicomponent, dialogue-based and tailored interventions appear to be more effective.[36] In addition, an important aspect of the Swedish national immunisation programme is the Child Healthcare Services structure and its responsibility for providing and implementing childhood vaccinations.[37] Parents often meet the same nurses for the child's health and development checkups as well as for vaccinations, and therefore, the nurses have a crucial role in the success of the vaccination programme. Although our present study suggests improvements in the dialogue from the perspective of the nurses, the nurses will still be faced with recurring challenges and parents having questions and concerns regarding vaccinations. Hence, continued work and effort are needed to support the positive trend in this community. The current study assessed the dialogue and situation 2 years after the implementation of the intervention. Further evaluations of long-term outcomes are needed as well as an assessment of the vaccination coverage in the community over time. As there are workforce turnover, it would be of importance that newly recruited nurses in the community are offered similar training and introduction to the tools. The current intervention was implemented in 2015–2017, repeating the seminars and reinforcing the information for all nurses may also be important for continued capacity building and sustainability. Generally, vaccinating nurses could benefit from training and boosting of knowledge routinely regarding communication with parents about vaccinations.

Evaluations of tailored interventions are needed to increase the evidence base for effective interventions that can be used by public health authorities to increase vaccination acceptance or uptake and contribute to resilience for national immunisation programmes. By addressing pockets with suboptimal vaccination coverage and identifying the facilitators and barriers to vaccination, tailored interventions can be designed and implemented to improve vaccination uptake and health equity.

### Methodological considerations

Throughout the research process, trustworthiness has been strengthened by several activities. Several members of the interdisciplinary research team had expertise in qualitative methodology and were also familiar with the setting, community and culture over an extended period before the intervention was implemented. Moreover, the research team familiarised themselves with the data initially and regularly discussed the analysis to strengthen its validity. This study is limited, however, as it only focused on the perspective of the nurses; further studies are needed to capture the perspective of the parents. Although the number of participants in this interview study is small, the results are relevant for this specific community and setting, generalising the findings to other settings should be done with caution.

### CONCLUSION

Nurses' perceived the dialogue to be facilitated and enhanced with parents' vaccination concerns, following the implementation of tailored interventions. Particularly, nurses increased confidence in conveying evidence-based information about autism and MMR vaccination and felt better equipped to convey this information to the parents and to dispel the myth linking autism and MMR in their dialogue with the parents. In addition, the tailored tools were also perceived to reduce language barriers. Child health nurses have a crucial role in the vaccine acceptance and uptake in the Swedish national immunisation programme, and interventions aiming to strengthen the nurses in their communication and dialogue with parents are therefore important. Further studies are needed for assessing the long-term outcomes of the multicomponent intervention as well as vaccination coverage within the studied community.

**Author affiliations**
[1]Department of Public Health Analysis and Data Management, Public Health Agency of Sweden, Solna, Sweden
[2]Department of Clinical Microbiology, Department of Translational Medicine, Lund University, Lund, Sweden
[3]Department of Global Public Health, Karolinska Institutet, Stockholm, Sweden
[4]Department of Immunization, Vaccines and Biologicals, Unit Essential Programme on Immunization, WHO, Geneve, Switzerland
[5]The Office of the Head for Communicable Disease Control and Health Protection, Public Health Agency of Sweden, Solna, Sweden

**Acknowledgements** The authors wish to thank the Stockholm Region child health services for their support and collaboration in this study. A sincere thank you to all the nurses at the child healthcare centres who kindly participated with openness and commitment.

**Contributors** EA, AJ, AK, AL and KG-R designed the study. All authors contributed to the analysis and interpretation of data. EA, AJ and KG-R drafted the first version of this paper, and AK, AL and AR have revised the paper critically and substantially. The final version of the paper has been approved by all authors, and they all agree to be accountable for all aspects of the work. EA and KG-R are guarantor.

**Funding** The current study was funded by The Public Health Agency of Sweden.

**Competing interests** None declared.

**Patient and public involvement** Patients and/or the public were not involved in the design, or conduct, or reporting, or dissemination plans of this research.

**Patient consent for publication** Not applicable.

**Ethics approval** The study received ethical approval for the interviews from the Regional Ethics Committee in Stockholm, Sweden, Dnr 2016/1518-31/5. Participants gave informed consent to participate in the study before taking part.

**Provenance and peer review** Not commissioned; externally peer reviewed.

**Data availability statement** No data are available.

**ORCID iD**
Emma Appelqvist http://orcid.org/0000-0002-7058-2419

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
