## [Reviewer comments · BMJ Open]

ARTICLE DETAILS

TITLE (PROVISIONAL)	Exploring nurses' experiences of a tailored intervention to increase MMR vaccine acceptance in a Somali community in Stockholm, Sweden: a qualitative interview study
AUTHORS	Appelqvist, Emma; Jama, Asha; Kulane, Asli; Roth, Adam; Lindstrand, Ann; Godoy-Ramirez, Karina

VERSION 1 – REVIEW

REVIEWER	Marie Hill University of London, Nursing Division
REVIEW RETURNED	31-Aug-2022

GENERAL COMMENTS	Abstract Objectives Line 22. Apostrophe change. nurses', instead of nurse's. Conclusion Lines 50 – 51. Suggest the following sentence to be re phrased The tailored interventions targeting nurses facilitated and improved the nurses encounters and dialogue with parents who had MMR vaccine concerns. Lines 51 – 56 The first sentence (i.e., Lines 51 – 54) is too long. Suggest two sentences to convey a clearer meaning. For example, "dispel the myth" needs unpicking. What was this myth that nurses dispelled for these vaccine hesitant parents? Strengths and limitations of this study First bullet point. Line 6. Apostrophe change. nurses', instead of nurse's. Second bullet point. Line 9. Suggest the following: In depth interviews of nurses provided unique insights into the experiences in dealing with vaccine hesitant parents. Introduction Lines 5 and 15. Change percentage to %. Line 8. All cases are either imported or epidemiologically linked to the imported cases. A reference is required to support this sentence. Lines 15 – 16. With the accumulation of susceptible unimmunized populations, the area had an increased risk of outbreaks. Specify what disease outbreaks you are referring to. Line 52. Apostrophe change. nurses', instead of nurse's. N.B. Check all reference to "nurse's" throughout this article to ensure the apostrophe is correctly positioned. Methods Summary of the TIP-project in Rinkeby and Tensta, Stockholm
--

	Line 11. Apostrophe change. nurses', instead of nurse's Lines 13 – 14. Can you confirm how many nurses at CHC were approached initially? It would seem one nurse declined. Were any reasons provided by this nurse? Lines 14 – 16. Suggest the following: The overall design and implementation of the intervention for both parents and nurses has been described in previously published research (20). Lines 19 – 20. Suggest the following. During three consecutive weeks in August and September 2015, all nurses were invited to a series of seminars. Lines 36- 37. Suggest writing this sentence as two separate sentences. A two sided card was provided to nurses to use in their consultations and to share with parents. This card consisted of information in the Somali language on one side with the same information on the other side of the card in Swedish. Data collection Line 60 (page 4) and line 3 (page 5). What were the minor adaptations made? Results Lines 38 – 39. Suggest the following rephrasing: The results are presented in detail below with the overarching theme in bold and the subthemes in italics and underlined. Line 59. Delete "it". Discussion Line 18. Apostrophe change. parents', instead of parent's. Lines 33 – 34. Provide data on measles cases in Europe in 2018 – 2019, with a supporting reference. Line 34. Several of the countries.....Again, more detail required (i.e., the countries names) with a supporting reference. Conclusion Lines 33 – 35. I would suggest re phrasing this sentence to convey a clearer meaning. Overall There are a number of occasions where the writers refer to parents as parents having low vaccine acceptance. In these instances, consider the use of vaccine hesitant parents. It would have been beneficial to have detail of the 11 nurse characteristics in terms of:  • Gender • Academic levels (i.e., highest academic level attained) • Ethnicity • Employment status (i.e., whether full or part time) • Years in clinical practice as a nurse • Immunization training • Latter immunization update. A recommendation to consider would be for nurses working in public health (and facilitating the delivery of immunization programs) to have mandatory immunization training, with annual updates.
--	--

REVIEWER	Akhenaten Tankwanchi University of Washington School of Public Health
REVIEW RETURNED	02-Nov-2022

GENERAL COMMENTS	I am familiar with the research team's previous work on the same topic and believe that their contribution to this field of inquiry is
--

	valuable. This latest study is also informative and could be a valuable addition to the growing body of research on vaccine hesitancy in high-risk subpopulations like immigrants and other minoritized ethnic groups. However, the manuscript must be improved before it can be published in any credible journals like BMJ Open. Two major concerns are listed below while additional detailed comments are embedded in the attached PDF manuscript file. 1. Lack of a clear theoretical framework to guide the methods and results. The authors mentioned the "TIP approach" and "situational analysis" but did not specify what these entails, how these structure their study, and the theoretical underpinnings that support them. Merely stating that they are based on social and behavioral sciences is not very informative. 2. Insufficient depth of the analysis. The authors claimed that they conducted "in-depth interviews" from which they gleaned out "thorough and rich data." However, the depth of their data and attendant analysis is wanting and must be developed further. It is notable that the research team conducted the interviews since 2017. Going by their own estimates, they collected about 375 minutes worth of interview data with 11 nurses, which may translate into 100 plus pages of interview transcripts. Yet, they claimed that only one main theme and two subthemes emerged from coding the data. No discussion of data saturation is provided, while some important findings are merely mentioned in passing (e.g.: "A few nurses recalled and gave examples of situations where parents with low acceptance had changed their mind after watching the movie and chose to vaccinate their child against MMR.") The reviewer provided a marked copy with additional comments. Please contact the publisher for full details.
--	---

REVIEWER	Dell Horey La Trobe University, Public Health
REVIEW RETURNED	06-Nov-2022

GENERAL COMMENTS	This is an interesting - and potentially useful paper. However there are some issues that need to be clarified before the paper will be suitable for publication. Most of the issues that I have identified should be relatively easy to address. I recommend that the authors refer to a standardised reporting checklist for qualitative research. There are two that would be suitable: "Standards for Reporting Qualitative Research (SRQR)" O'Brien, B. C., Harris, I. B., Beckman, T. J., Reed, D. A., & Cook, D. A. (2014). Standards for reporting qualitative research: a synthesis of recommendations. Academic Medicine, 89(9), 1245-1251. Or the "Consolidated criteria for reporting qualitative research (COREQ)" Tong A, Sainsbury P, Craig J. Consolidated criteria for reporting qualitative research (COREQ): a 32-item checklist for interviews and focus groups. Int J Qual Health Care. 2007;19(6):349-357. There are a number of unsupported or inconsistent statements that need to be resolved. * Method - Interview guide
--

Page 4 (Lines 28-30) state “At the end of the third and final seminar, nurses attending the seminar (n=12) were asked to participate in a short survey with 11 questions for evaluation purposes. The result of the survey was used to inform the interview guide”, but later on the same page (Lines 55-58) “The interviews were semi-structured using a thematic guide. The outline was adapted from a previous study. It also included questions regarding experiences of the interventions, benefits, and changes in daily work since the intervention as well as the perception of parents’ experiences of the interventions.”

* Results and Discussion

Some of these may be translation issues that need to be reviewed as the phrasing raises concerns that the stated findings are unsupported by the evidence provided..

Page 6 (Lines 58-59)

It is not initially clear that the statement “Instead of engaging in a dialogue in such a situation, some nurses just offered the possibility to postpone the MMR vaccination” refers to behaviours before the intervention.

Page 9 Discussion (Lines 14-15)

“The tailored intervention supported the dialogue, dispelled myth and reduced language barriers while also gave nurses a better understanding of the parental perspective and cultural insights concerning MMR vaccination and reasons for the parents’ concerns.” There is nothing reported in the results about the nurses gaining “a better understanding of the parental perspective”. If the authors infer this from their findings this should be clear and not reported as an actual result.

Page 9 (Lines 55-60)

“Although our present study suggests improvements in the dialogue, the nurses will still be faced with recurring challenges and parents having questions and concerns regarding vaccinations. Hence, continued work and effort are needed to support the positive trend in this community. The current study assessed the dialogue and situation 2 years after the implementation of the intervention. Further evaluations of long-term outcomes are needed as well as assessment of the vaccination coverage in the community over time”

It is not clear that the dialogue was “assessed” rather the authors are reporting perceptions of the dialogue – there is no independent assessment of its quality.

Page 10 (Lines 3-6)

“Based on the results of our current study, it would be important that newly recruited nurses in the community are offered similar training and introduction to the tools. The current intervention was implemented 2015-2017, repeating the seminars and reinforcing the information for all nurses may also be important for continued capacity building and sustainability.”

While I agree that the issues in the study highlight sustainability as a problem, but this is not due to any evidence provided in the results. Rather, the issue was highlighted by a process in the method that was amended to accommodate the significant proportion of the workforce (25%) that had changed and so had not attended the seminar. This meant that questions about the seminar could not be asked of them. In addition, the views on the use of the card and film by these participants were not differentiated from others, so it is

	unclear whether access to the film and cards were sufficient to improve nurse knowledge. Page 10, (Lines 34-35) Conclusion “the nurse’s dialogue was facilitated and enhanced with parents’ vaccination concerns” It would be more correct to say that this about the nurses’ perceptions as the actual dialogue with the parents was not monitored in this study. *Structure There are couple of sections in the paper that would appear to be misplaced. Suggest that the section with the sub-heading on Page 5 “Summary of the TIP-project in Rinkeby and Tensta, Stockholm” should be included in the Background not Methods, as this does not describe the methods for the present study. The following paragraph [Page 6 (Lines 38-40)] “The results are presented in detail below with theme as bold and subthemes as italic and underlined headings. Important quotes from the interviews have been chosen to emphasize and support the results” should be included in Methods section not Results. The paragraph beginning “Globally, there was an increase of measles cases in Europe in 2018-2019” [Page 9 Discussion (Lines 33-41)] would be better placed in the Background to the study, as it sets the context of the study but does not discuss the study findings. * Analysis – Framing of themes The wording of the themes is rather “clunky” which makes them uneasy to communicate, and I suggest that they are reconsidered so that the findings can be more easily disseminated. For example “Facilitation and improvement of nurses' encounter and dialogue regarding MMR vaccination with parents having vaccine concerns” does not actually describe the data that is reported. Suggest something like “Perceptions of improved responses to parent vaccination concerns” and/or “Reluctance to persuade parents rather than enable their choices” “Increased knowledge and confidence in conveying evidence-based information about autism and MMR vaccination” could be something like “knowledge builds confidence to convey vaccine information” “Specific usefulness of the tools to facilitate dialogue, dispel myth and reduce language barriers” This does not seem like an actual theme – and I suggest that rather than report on the individual components that the authors consider how these tools were used. For example, there does seem to be some variation in how the film was used and some of the quotes used seem to indicated that the integration of the tools – the film and card – was important, particularly the use of Somali language and parents’ voices. *Check English expression For example, Page 7 (Lines 33-35) “Particularly, the nurses remembered the seminar about the epidemiological situation of measles in Somalia, the Somali vaccination program, and how parents in and from Somalia reason regarding vaccinations”
--	--

	(Lines 58-59) “and the complexity of factors potentially involved in the onset of disease, despite the etiology not yet being unclear.” Assume that the authors mean “clear” here. Thank you
--	---

VERSION 1 – AUTHOR RESPONSE

Reviewer 1:

Comments to the Author: An interesting study. My comments are included in the uploaded file.

Abstract

Objectives

Line 22. Apostrophe change. nurses', instead of nurse's.

- Adjusted as suggested

Conclusion

Lines 50 – 51.

Suggest the following sentence to be re phrased

The tailored interventions targeting nurses facilitated and improved the nurses encounters and dialogue with parents who had MMR vaccine concerns.

- Thank you for your suggestion. We have revised the sentence.

Lines 51 – 56

The first sentence (i.e., Lines 51 – 54) is too long. Suggest two sentences to convey a clearer meaning. For example, “dispel the myth” needs unpicking. What was this myth that nurses dispelled for these vaccine hesitant parents?

- Thank you for your suggestion. We have revised and deleted the sentence based on feedback from another reviewer.

Strengths and limitations of this study

First bullet point. Line 6. Apostrophe change. nurses', instead of nurse's.

- Adjusted as suggested

Second bullet point. Line 9. Suggest the following:

In depth interviews of nurses provided unique insights into the experiences in dealing with vaccine hesitant parents.

- Adjusted almost as suggested with slight changes to the sentence.

Introduction

Lines 5 and 15. Change percentage to %.

- Adjusted as suggested

Line 8. *All cases are either imported or epidemiologically linked to the imported cases.* A reference is required to support this sentence.

- We have revised and added a reference to support the sentence. “All recent cases have either been imported or epidemiologically linked to an imported case, apart from a few sporadic cases of unknown origin (personal communication, H el ene Englund, The Public Health Agency of Sweden, 2022-11-15)”

Lines 15 – 16. *With the accumulation of susceptible unimmunized populations, the area had an increased risk of outbreaks.* Specify what disease outbreaks you are referring to.

- We have adjusted and specified the risk of measles

Line 52. Apostrophe change. nurses', instead of nurse's. N.B. Check all reference to “nurse’s” throughout this article to ensure the apostrophe is correctly positioned.

- Thank you for pointing this out, we have read through the manuscript and made adjustments throughout.

Methods

Summary of the TIP-project in Rinkeby and Tensta, Stockholm

Line 11. Apostrophe change. nurses', instead of nurse's

- Adjusted as suggested

Lines 13 – 14. Can you confirm how many nurses at CHC were approached initially? It would seem one nurse declined. Were any reasons provided by this nurse?

- There were 12 nurses working at the CHCs initially. One nurse declined to participate due to being on leave.

Lines 14 – 16. Suggest the following:

The overall design and implementation of the intervention for both parents and nurses has been described in previously published research (20).

- Adjusted as suggested. The reference is also updated as the article is now published.

Lines 19 – 20. Suggest the following.

During three consecutive weeks in August and September 2015, all nurses were invited to a series of seminars.

- Adjusted as suggested

Lines 36- 37. Suggest writing this sentence as two separate sentences.

A two sided card was provided to nurses to use in their consultations and to share with parents. This card consisted of information in the Somali language on one side with the same information on the other side of the card in Swedish.

- Adjusted as suggested

Data collection

Line 60 (page 4) and line 3 (page 5). What were the minor adaptations made?

- Thank you for the relevant question. We have revised the sentence in the manuscript to be more specific regarding what changes was made. The changes basically additions of more probing questions and reorganization of a few questions to get a better flow in the interview.

Results

Lines 38 – 39. Suggest the following rephrasing:

The results are presented in detail below with the overarching theme in bold and the subthemes in italics and underlined.

- Thank you for your suggestion. We have taken these sentences out and instead added a new sentence in the methods section regarding the quotes.

Line 59. Delete "it".

- Adjusted as suggested

Discussion

Line 18. Apostrophe change. parents', instead of parent's.

- Adjusted as suggested

Lines 33 – 34. Provide data on measles cases in Europe in 2018 – 2019, with a supporting reference.

- Thank you for your suggestion. We have added a reference and added more details in the text.

Line 34. *Several of the countries.....* Again, more detail required (i.e., the countries names) with a supporting reference.

- Thank you for the suggestion. We have revised the sentence to be more specific. Although we didn't add a list of countries as it would be too long for the sentence.

Conclusion

Lines 33 – 35. I would suggest re phrasing this sentence to convey a clearer

meaning.

- Thank you for your suggestion, we have revised the sentence to provide a clearer meaning.

Overall

There are a number of occasions where the writers refer to parents as *parents having low vaccine acceptance*. In these instances, consider the use of *vaccine hesitant parents*.

- We have intentionally chosen to refer to the parents in this article as having lower vaccine acceptance instead of using vaccine hesitant parents. This is due to that we want the term to also include parents having concerns but still vaccinate and not just delay or reject vaccination. We perceive the term vaccine acceptance to more including of all parents and their stance in vaccination.

It would have been beneficial to have detail of the 11 nurse characteristics in terms of:

- Gender
 - Academic levels (i.e., highest academic level attained)
 - Ethnicity
 - Employment status (i.e., whether full or part time)
 - Years in clinical practice as a nurse
 - Immunization training
 - Latter immunization update.
-
- We have added more information regarding gender, academic levels and years of clinical practice as a child nurse in the article. However, at the recommendation of the editor, we have not included it as a table but rather summarized the information in order to protect the participants from being identified.

A recommendation to consider would be for nurses working in public health (and facilitating the delivery of immunization programs) to have mandatory immunization training, with annual updates.

- Thank you, this is a valuable point to add. We have included a sentence in the discussion.

Reviewer 2:

Comments to the Author:

I am familiar with the research team's previous work on the same topic and believe that their contribution to this field of inquiry is valuable. This latest study is also informative and could be a valuable addition to the growing body of research on vaccine hesitancy in high-risk subpopulations like immigrants and other minoritized ethnic groups. However, the manuscript must be improved before it can be published in any credible journals like BMJ Open.

Two major concerns are listed below while additional detailed comments are embedded in the attached PDF manuscript file.

1. Lack of a clear theoretical framework to guide the methods and results. The authors mentioned the "TIP approach" and "situational analysis" but did not specify what these entails, how these structure their study, and the theoretical underpinnings that support them. Merely stating that they are based on social and behavioral sciences is not very informative.
 - Thank you for raising an important question. The tailoring immunization approach (TIP) developed by WHO is based on social and behavioral science but also underpinned by theoretical frameworks, the Capability-Opportunity-Motivation-Behaviour (COM-B) model and the Behaviour Change Wheel (BCW) framework. We have expanded the paragraph where we describe the TIP approach to include more details and to show its theoretical support. Now that we revised the manuscript we also changed the wording from situational analysis to "formative research" to match the wording of the latest version of the TIP-guide. The TIP-guide informed the project overall, for which this article is just a piece of several studies as part of the evaluation of our intervention project.

What is Situation Analysis? Is this supposed to be the framework that structures your study? If so, can you a) tell the readers what it entails? b) organize both your Methods and Results sections according to this framework? If not, can you provide a framework for your study?

- Please see previous response regarding the TIP approach and its framework. We have revised and included more information regarding the TIP approach and what it entails. This manuscript is based on our intervention and therefore we cannot organize the methods and results according to the TIP approach, as the approach includes a lot more than just the evaluation. This is one of many studies included for the project. For this manuscript, we have used thematic analysis and methodology for guidance as we had a predetermined focus for our study to evaluate the tools implemented for nurses.

2. Insufficient depth of the analysis. The authors claimed that they conducted "in-depth interviews" from which they gleaned out "thorough and rich data." However, the depth of their data and attendant analysis is wanting and must be developed further. It is notable that the research team conducted the interviews since 2017. Going by their own estimates, they collected about 375 minutes worth of interview data with 11 nurses, which may translate into 100 plus pages of interview transcripts. Yet, they claimed that only one main theme and two subthemes emerged from coding the data. No discussion of data saturation is provided, while some important findings are merely mentioned in passing (e.g.: "A few nurses recalled and gave examples of situations where parents with low acceptance had changed their mind after watching the movie and chose to vaccinate their child against MMR.")

- Thank you for your feedback as to what we can develop further to improve the manuscript. In terms of the expression of "in-depth" interviews, we have followed quantitative methodology for conducting interviews, using probing questions throughout and therefore we view them as in-depth interviews. We also perceive rich data in terms of the objective for this study which was to explore the perception of the intervention by the nurses. In terms of saturation of the data, all nurses employed at the health care centers involved in the intervention were invited for the study. All but one who was on leave, participated. Hence we interviewed all eligible participants.

Title: of a tailored intervention?

- Adjusted as suggested

Abstract

Since you are already introducing sample size and participants here, consider merging this subsection with the Participants subsection to provide better context, avoid redundancy, and improve the fluidity of your writing.

- Thank you for your suggestion. We have revised the majority of the abstract to improve the text further.

Given that this study was conducted five years ago, it may be more appropriate to use the past tense here. "This study was part of a multi-component intervention..."

- Adjusted as suggested

Suggested revision:

"This study was part of a Sweden-based 2015-2017 multi-component intervention targeting Somali parents and the nurses who work with them in the Rinkeby and Tensta neighborhoods of Stockholm, areas with documented MMR under-vaccination.

- Thank you for your suggestion. We have revised and made slight revisions to the suggested sentence.

"Nurses at child health centers were involved in a tailored intervention implemented in 2015 that included a series of seminars..."

- Thank you for your suggestion. We have revised and made slight revisions to the suggested sentence.

Please refine this section. Remove italics and quotation marks as these can suggest verbatim quotations from respondents.

- Thank you for your suggestion. We have revised and removed both the italics and quotation marks.

This entry reads like the Results section.

- Thank you for the suggestion, we have revised the section.

Strengths and limitations of this study

enabled understanding of [Swedish] nurses' experiences of an intervention...

- Thank you for your suggestion, we have revised as suggested.

The verbatim quotations provided in the text do not sufficiently reflect thoroughness and richness of data.

- Thank you for your suggestion. We do not agree however with the statement, as we see our data as rich. The sentence has been revised as suggested by another reviewer.

Introduction

Given that this study was conducted five years ago, it may be more appropriate to use the past tense here. "This study was part of a multi-component intervention..."

- Adjusted as suggested

Suggested revision:

"This study was part of a Sweden-based 2015-2017 multi-component intervention targeting Somali parents and the nurses who work with them in the Rinkeby and Tensta neighborhoods of Stockholm, areas with documented MMR under-vaccination.

- Adjusted as suggested

"Nurses at child health centers were involved in a tailored intervention implemented in 2015 that included a series of seminars..."

- Adjusted as suggested

Strengths and limitations of this study

enabled understanding of [Swedish] nurses' experiences of an intervention...

- Adjusted as suggested

Introduction

Line 4: redundant

- Adjusted

Suggested revision: The measles, mumps, rubella (MMR) vaccination coverage has been high and stable throughout the last decade in Sweden, where 97.1 percent of two-year children were vaccinated against MMR in 2020.

- Adjusted as suggested

which afflicted the Somali community

- Adjusted as suggested

children of Somali descent?

- Thank you, we agree this would be interested to know. Unfortunately, neither country of origin nor ethnicity for cases are included in the national statistics and therefore impossible to know.

Where is the supporting evidence or reference for this specific statement?

- A relevant reference has been added

Could you also briefly provide a picture of measles coverage in Somalia? Data for the GBD 2019 suggest that Somalia has the second highest rate of death by measles in Sub-Saharan Africa (after Niger). See: <https://vizhub.healthdata.org/gbd-compare/>

- Thank you for your suggestion. We were discussing this as we wrote the manuscript and in the end we decided not to include statistics from neither Africa nor Somalia as we believe the scope and target groups differs. We chose to include examples from Sweden and Minnesota as the context are more similar and the focus is on immigrants of Somali descent.

Consider using this phrase in your title instead of the plural form.

- Adjusted as suggested

Methods

The paper listed in the References suggests that it has not yet been published (although it has been accepted). In order to avoid any anachronism, I suggest that the authors substitute the adverb "previously" with "elsewhere".

- The reference has now been published and the reference list has been updated. We have revised according to the suggestion by reviewer 1.

The word "card" in this phrase is not self-evident and needs additional details. Are these playing cards, illustrative cards with pictographs? Please explain.

Also, given the intervention's reliance on visuals (film and card), and the requirement to sufficiently describe the methods to enable replicability, it could be helpful to include the picture of one of the visuals or didactic materials used in the intervention.

- Thank you for raising this. To help the reader understand the card better, we have tried to clarify the description of the card in the manuscript. We have also added a reference to a website to where both the card and film has been published so it is possible to get a full visual of both intervention tools.

If this film is still available on the YouTube website, please kindly provide a reference and a link to it in the References section so that the readers may access it. If this film is inaccessible to readers, please explain why in the appropriate space of the manuscript submission portal (data accessibility).

- The film is no longer on YouTube, but still accessible online. We have added a reference to the website where the film is published and can be viewed.

This addresses my comment above. Given the description provided, it may be more accurate to call it a "factsheet" in lieu of card.

- Thank you for relevant suggestions, however, we still feel that information card best describes the tool. We have also already published an article where the information card is describes, so it would be confusing to readers to change the name at this point.

As suggested previously about the YouTube link, please provide a reference and the active link to this website.

- We have added a reference to the website where the film is published.

"...facilitated the invitation and recruitment into the study."

- Adjusted as suggested

Suggested revision: "Three nurses employed after the 2015 seminar series were briefed about the seminars and given the materials (film and factsheet) used in the intervention. Hence, they were deemed eligible for the study and were invited to participate in the interviews."

- Adjusted as suggested

Should we conjecture that one nurse declined to participate? Please explain the discrepancy between the number of invitees (n=12) and the number of interviewees (n=11).

- We revised to clarify that 12 nurses were invited to take part in the study and 11 agreed to participate. One nurse declined due to being on leave.

Please include this interview guide in your methods section (or as an appendix). If it is too long, highlight a representative sample of the main questions.

- Thank you for your suggestion. We will include our interview guide in the appendix. The original interview guide was written in Swedish but we have translated it to English for the purpose of submission as appendix to our manuscript.

Suggested revision: "The three nurses who began working at the local CHC after the seminar series were only..."

- Adjusted as suggested

So the entire sample comprised only nurses who were Swedish natives and no Somalis? If this is the case, you should discuss this in the limitations.

- As the study participants are limited to those employed at the CHCs included in the intervention sites at the time of the study, we chose to conduct the interviews in Swedish as it's the official working languages to the nurses. We don't see the chosen language as a limitation as the nurses were all fluent in Swedish, and there was no language barriers for the interviews. We didn't collect any data regarding native backgrounds in order to prevent identification of participants, but native backgrounds were not only Swedish.

Instead of merely directing the readers to a handbook about thematic analysis in psychology, you should first briefly describe what a thematic approach entails.

- Thank you for the suggestion. We have revised the data analysis paragraph in the methods section to include more description of our chosen analysis for the thematic analysis.

The interview transcripts? Consider using the active form here and throughout your narration: "The first author read the interview transcripts several times to become..."

- Adjusted as suggested

To give more credibility to your research, you may consider including (in your Results or Methods) a screenshot of your Excel worksheet with granular/verbatim data and associated codes.

- Thank you for your suggestion. We conducted the analysis both manually and in Microsoft Office Excel. Please see the example that we have added in the methods section.

Results

Delete or move the Methods section.

- The sentence has been deleted. The information is included in the method-section instead.

Given that increasing vaccine acceptance among Somali parents was the ultimate goal of the intervention, this is arguably the most important finding of your study and should be further developed. Instead of saying "a few nurses recalled", please give the specific number of nurses who reported that parents reconsidered their decision against MMR vaccination and give the specific number of parents who actually had their children vaccinated against measles post-intervention.

- Thank you for your suggestion, it's a valid point to make. During the interviews five nurses recalled parents who have decided to vaccinate after watching the film. We have also revised the sentence and included this information. Our results reflects what the nurses have said

during the interviews and as the nurses could not recall an exact number of parents, we cannot provide more details in our results. Hence, it is impossible to know how many parents have reconsidered their decision after the intervention. Therefore, we are conducting a separate retrospective, registry based study to assess the vaccination status of the children attending the CHCs. The registry-based study is ongoing and will be published separately.

Can you briefly elaborate on the expertise of these trusted Somali community members? Were there any healthcare providers among them?

- Thank you for pointing this out. The film and the participating experts are described earlier in the manuscript, in section "Summary of the TIP-project in Rinkeby and Tensta, Stockholm". The experts include health care professional, religious community leader and parents. The participants and the content of the film is also further described in a reference included in our manuscript: Jama A, Appelqvist E, Kulane A, Karregård S, Rubin J, Nejat S, et al. Design and implementation of tailored intervention to increase vaccine acceptance in a Somali community in Stockholm, Sweden - based on the Tailoring Immunization Programmes approach. *Public Health in Practice*. 2022;4:100305.

If another measles outbreak occurred in this same area post intervention, what does this suggest about the effectiveness of this intervention? It will be helpful to discuss this in the Discussion section.

- Thanks for the suggestion. Please see the point above on a separate registry-based study on vaccination status. We prefer to discuss this point in conjunction with that study. No changes done to the manuscript due to this comment.

Discussion

Suggested revision:

"Findings from the present study suggest that tailored interventions with culturally appropriate didactic materials can facilitate vaccine-related communication with and ultimately improve vaccine acceptance among vaccine-hesitant parents of Somali descent..."

- Thank you for your suggestion. We have revised to include a slightly rephrased version of the suggested sentence.

"more assertive"?

- Thank you for your suggestion. However, we don't feel like assertive is a better word to describe the sentence and therefore we haven't revised the sentence.

As commented earlier, please explain why despite this vaccine acceptance intervention implemented in 2015 there was a measles outbreak in the area in 2017.

- Thank you for this relevant question. Behavioral changes for individuals take time and we didn't expect the vaccination coverage rise immediately. As the implementation of the intervention started in 2015, the focus was to address MPR-vaccinations offered at 18 months. Over the years there are older cohorts of children who are not vaccinated, but catch-up vaccinations were not part of our intervention design. Hence, there are potentially many reasons for why an outbreak of measles occurred in 2017.

Similar barriers to vaccination...

- Adjusted as suggested

Consider providing the recommended vaccination coverage threshold for achieving measles herd immunity.

- Thank you, we have added details about the threshold as suggested.

"...a low or suboptimal vaccination..."

- Revised to just state suboptimal coverage

is it necessary to write both words? what's the difference?

- Adjusted and just kept "crucial role" checkups?

- Adjusted as suggested

Why did it take so long to publish your findings?

- The study was conducted 2 years after the intervention as we wanted to allow time for the nurses to use the tools during a longer period of time. Unfortunately, the manuscript was not finalized before the covid-19 pandemic hit and the last few years the authors have been involved with the pandemic response full time until recently.

Do you make a difference between "vaccine acceptance" and "vaccine uptake"? If yes, briefly describe or discuss this difference. If not, use only one term or use the conjunction "or" instead of "and".

- Thank you for asking this. We do differentiate between vaccine acceptance and vaccine uptake as they relate to different aspects. In our view, vaccine uptake is related to the actual vaccination coverage/status while vaccine acceptance is not necessarily actual behavior but reflects the degree the vaccination is accepted, questioned or postponed, which depends on many factors that influences vaccine decisions.

Methodological considerations

Strengths and limitations?

- Thank you for your suggestion, we are open to use this heading as well but as the section with bullet points following the abstract has the same heading, we chose to keep the methodological considerations as subheading in order not to make it confusing for the reader.

Suggestion: "Trust between the nurses and the local Somali community of Ribenky and Tensta has been strengthened as a result of the activities conducted during this tailored intervention."

- Thank you for providing a suggestion for revision of the sentence. The meaning of the suggestion is not in line with the point that we were trying to make. Our focus for this sentence is for the trustworthiness of the study from a qualitative methodological perspective. Hence, we haven't revised as suggested.

This reads like an entry from the Methods section. Consider deleting or moving it to that section.

- Thank you for your suggestion. We moved the sentence to the data collection heading in the methods section.

Suggestion: "Several members of our interdisciplinary research team have become more knowledgeable of the local Somali community and more culturally competent to work with them."

- Thank you for your suggestion. Although your suggestion is accurate, it seems like our intention behind the sentence was not entirely clear and therefore we have revised the sentence. We intended to highlight that several of the members of the research team were familiar with the setting, community and culture even before the study started and not only gained more knowledge during the study.

"...discussed the analysis to strengthened its validity."

- Thank you for your suggestion. We have revised as suggested.

Do the specific information provided by these three nurses suggest that they were less knowledgeable (and confident) than their counterparts about the relevant vaccine issues and how to address them? If not, consider deleting these two sentences.

- Thank you for highlighting this. We don't perceive different between the nurses and therefore we have deleted the sentences as suggested.

What counts as a long-term outcome? Are the outcomes reported in the Results and the Methodological Considerations sections considered short- or medium-term? Seven years have elapsed since this intervention study was conducted, and it has been five years since the present evaluation was conducted. While informative, this study would have been more valuable if the authors had supplemented their 2017 interview data with fresher data from more recent studies and even

informal personal/email communication with the nurses that participated in their study. As regards long-term outcome, assuming there has been no additional measles outbreaks in Ribenky & Tensta since 2017, can one speculate that this may be due in part or in whole to the TIP intervention?

- Thank you for your question, it's an important point to discuss. This manuscript is just one of several studies ongoing to evaluate the intervention. To assess the long-term outcomes and the actual differences in vaccination coverage in the area, registry studies are being conducted. Since 2017, there hasn't been any more outbreaks in the area. Since 2020 there have been unusually few cases of measles in Sweden due to the restrictions implemented to prevent the spread of covid-19 and therefore it's difficult to speculate what impact there might be due to our intervention specifically.

Reviewer 3:

Comments to the Author:

This is an interesting - and potentially useful paper. However there are some issues that need to be clarified before the paper will be suitable for publication. Most of the issues that I have identified should be relatively easy to address.

I recommend that the authors refer to a standardised reporting checklist for qualitative research. There are two that would be suitable:

- "Standards for Reporting Qualitative Research (SRQR)" O'Brien, B. C., Harris, I. B., Beckman, T. J., Reed, D. A., & Cook, D. A. (2014). Standards for reporting qualitative research: a synthesis of recommendations. *Academic Medicine*, 89(9), 1245-1251.
- "Consolidated criteria for reporting qualitative research (COREQ)" Tong A, Sainsbury P, Craig J. Consolidated criteria for reporting qualitative research (COREQ): a 32-item checklist for interviews and focus groups. *Int J Qual Health Care*. 2007;19(6):349-357.

Thank you for the suggestion. As the editor also suggested to include the SRQR checklist we have we have provided a checklist for our re-submission.

There are a number of unsupported or inconsistent statements that need to be resolved.

* Method - Interview guide

Page 4 (Lines 28-30) state "At the end of the third and final seminar, nurses attending the seminar (n=12) were asked to participate in a short survey with 11 questions for evaluation purposes. The result of the survey was used to inform the interview guide", but later on the same page (Lines 55-58) "The interviews were semi-structured using a thematic guide. The outline was adapted from a previous study. It also included questions regarding experiences of the interventions, benefits, and changes in daily work since the intervention as well as the perception of parents' experiences of the interventions."

- Both the survey and a guide from a previous study was used to generate the semi-structured thematic guide that was used for the interviews. We have tried to clarify this in the methods section.

* Results and Discussion

Some of these may be translation issues that need to be reviewed as the phrasing raises concerns that the stated findings are unsupported by the evidence provided..

Page 6 (Lines 58-59)

It is not initially clear that the statement "Instead of engaging in a dialogue in such a situation, some nurses just offered the possibility to postpone the MMR vaccination" refers to behaviours before the intervention.

- Thank you for pointing this out. We have revised the sentence to clarify that the statements related to the period before the intervention as described by the participants: "Before the intervention some nurses described that, instead of engaging in a dialogue in such a situation, they just offered the possibility to postpone the MMR vaccination"

Page 9 Discussion (Lines 14-15)

"The tailored intervention supported the dialogue, dispelled myth and reduced language barriers while also gave nurses a better understanding of the parental perspective and cultural insights concerning MMR vaccination and reasons for the parents' concerns." There is nothing reported in the results

about the nurses gaining “a better understanding of the parental perspective”. If the authors infer this from their findings this should be clear and not reported as an actual result.

- Thank you for pointing this out. We have revised and tried to clarify our point in the discussion, as our line of thought was not clear for the readers. The parental perspective is related to the cultural insights, in gaining a better understanding from where the parents point of view in that sense.

Page 9 (Lines 55-60)

“Although our present study suggests improvements in the dialogue, the nurses will still be faced with recurring challenges and parents having questions and concerns regarding vaccinations. Hence, continued work and effort are needed to support the positive trend in this community. The current study assessed the dialogue and situation 2 years after the implementation of the intervention. Further evaluations of long-term outcomes are needed as well as assessment of the vaccination coverage in the community over time”

It is not clear that the dialogue was “assessed” rather the authors are reporting perceptions of the dialogue – there is no independent assessment of its quality.

- Thank you for pointing this out. It is true that we only explored the dialogue from the perspective of the nurses, hence we have tried to clarify the sentence to make it easier to understand.

Page 10 (Lines 3-6)

“Based on the results of our current study, it would be important that newly recruited nurses in the community are offered similar training and introduction to the tools. The current intervention was implemented 2015-2017, repeating the seminars and reinforcing the information for all nurses may also be important for continued capacity building and sustainability.”

While I agree that the issues in the study highlight sustainability as a problem, but this is not due to any evidence provided in the results. Rather, the issue was highlighted by a process in the method that was amended to accommodate the significant proportion of the workforce (25%) that had changed and so had not attended the seminar. This meant that questions about the seminar could not be asked of them. In addition, the views on the use of the card and film by these participants were not differentiated from others, so it is unclear whether access to the film and cards were sufficient to improve nurse knowledge.

- Thank you for pointing this out. We have revised the sentences and also taking another comment by one of the other reviewers into consideration.

Page 10, (Lines 34-35) Conclusion

“the nurse’s dialogue was facilitated and enhanced with parents’ vaccination concerns” It would be more correct to say that this about the nurses’ perceptions as the actual dialogue with the parents was not monitored in this study.

- Thank you for pointing this out. You’re correct that we only explored the experiences and perspective of the nurses, therefore we have revised the sentence to make it clear this was the case.

*Structure

There are couple of sections in the paper that would appear to be misplaced.

Suggest that the section with the sub-heading on Page 5 “Summary of the TIP-project in Rinkeby and Tensta, Stockholm” should be included in the Background not Methods, as this does not describe the methods for the present study.

- Thank you, we have revised as suggested and moved the paragraph to the background section. The description of the implemented tools for the nurses has been kept in the methods as we think it adds to the understanding of what has been implemented and therefore is most suited to be included in the methods section.

The following paragraph [Page 6 (Lines 38-40)] “The results are presented in detail below with theme as bold and subthemes as italic and underlined headings. Important quotes from the interviews have been chosen to emphasize and support the results” should be included in Methods section not Results.

- Thank you for the suggestion, we have deleted the sentences in the results section and included a slightly revised version of the sentences in the methods section.

The paragraph beginning “Globally, there was an increase of measles cases in Europe in 2018-2019” [Page 9 Discussion (Lines 33-41)] would be better placed in the Background to the study, as it sets the context of the study but does not discuss the study findings.

- Thank you for the suggestion. We have revised and moved the sentences to the beginning of the background section.

* Analysis – Framing of themes

The wording of the themes is rather “clunky” which makes them uneasy to communicate, and I suggest that they are reconsidered so that the findings can be more easily disseminated.

For example “Facilitation and improvement of nurses' encounter and dialogue regarding MMR vaccination with parents having vaccine concerns” does not actually describe the data that is reported. Suggest something like “Perceptions of improved responses to parent vaccination concerns” and/or “Reluctance to persuade parents rather than enable their choices”

“Increased knowledge and confidence in conveying evidence-based information about autism and MMR vaccination” could be something like “knowledge builds confidence to convey vaccine information”

“Specific usefulness of the tools to facilitate dialogue, dispel myth and reduce language barriers” This does not seem like an actual theme – and I suggest that rather than report on the individual components that the authors consider how these tools were used. For example, there does seem to be some variation in how the film was used and some of the quotes used seem to indicated that the integration of the tools – the film and card – was important, particularly the use of Somali language and parents' voices.

- Thank you for the feedback regarding the themes. As suggested, we have re-visited the final analysis and the author team has discussed the phrasing of the overall theme and the subthemes to make it easier to understand and communicate. Hence, the revised themes are:
 - perception of improved communication with parents
 - Feeling more confident to address parents MMR vaccine concerns
 - Diverse tools as useful support to dispel myth and reduce language barriers.
 - Tailored seminars
 - The information card
 - The film

*Check English expression

For example, Page 7 (Lines 33-35)

“Particularly, the nurses remembered the seminar about the epidemiological situation of measles in Somalia, the Somali vaccination program, and how parents in and from Somalia reason regarding vaccinations”

- Thank you for highlighting this sentence, we have revised as suggested.

(Lines 58-59) “and the complexity of factors potentially involved in the onset of disease, despite the etiology not yet being unclear.” Assume that the authors mean “clear” here.

- Yes, we have revised. Thank you for spotting our mistake.

VERSION 2 – REVIEW

REVIEWER	Marie Hill University of London, Nursing Division
REVIEW RETURNED	04-Jan-2023
GENERAL COMMENTS	Much improved and will add value to the body of evidence on supporting nurses in interacting with populations who are not vaccine accepting.

	While I have uploaded your final document with your detailed track changes, I have made minimal comments, with one minor change (i.e., relating to writing a number in full). The reviewer provided a marked copy with additional comments. Please contact the publisher for full details.
--	--

REVIEWER	Dell Horey La Trobe University, Public Health
REVIEW RETURNED	11-Jan-2023

GENERAL COMMENTS	Thank you for the opportunity to review this paper again and for addressing the issues previously raised. There are a few minor issues that still need attention to bring this paper to publishable standard. There are a several instances where the English expression isn't quite right. These may be addressed in the copy-editing stage but it would be helpful if the paper were reviewed carefully. For example, on Page 8 (Lines 6-7), in the sentence, "All interviews were conducted in a private space in the workplace, chosen by the interviewee to allow for openhearted discussions", "openhearted" is not the right word to use in this context. A better alternative would be "open discussions". Another example occurs on page 12 (Lines 12-13) "Generally, vaccinating nurses could benefit from training and boosting of knowledge routinely regarding vaccination communication." This statement is ambiguous as it could be interpreted as the nurses being vaccinated. The second statement in the introduction is unclear, "In Europe alone, 14 600 cases were reported in 2017 and in the WHO European Region more than 82 000 cases were reported [1, 2]". Is there an explanation for the difference in these two figures - are they both needed? Finally, the introduction is not well-structured, which makes it difficult to follow. Please see the attachment for a suggestion as to how this could be done more clearly. The reviewer provided a marked copy with additional comments. Please contact the publisher for full details.
---

VERSION 2 – AUTHOR RESPONSE

Reviewer: 1

Much improved and will add value to the body of evidence on supporting nurses in interacting with populations who are not vaccine accepting.

While I have uploaded your final document with your detailed track changes, I have made minimal comments, with one minor change (i.e., relating to writing a number in full).

- Thank you for your suggestion, we have adjusted as suggested.

Reviewer: 3

Thank you for the opportunity to review this paper again and for addressing the issues previously raised. There are a few minor issues that still need attention to bring this paper to publishable standard.

There are a several instances where the English expression isn't quite right. These may be addressed in the copy-editing stage but it would be helpful if the paper were reviewed carefully. For example, on

Page 8 (Lines 6-7), in the sentence, "All interviews were conducted in a private space in the workplace, chosen by the interviewee to allow for openhearted discussions", "openhearted" is not the right word to use in this context. A better alternative would be "open discussions".

- Thank you for the suggestion, the sentence has been revised accordingly.

Another example occurs on page 12 (Lines 12-13) "Generally, vaccinating nurses could benefit from training and boosting of knowledge routinely regarding vaccination communication." This statement is ambiguous as it could be interpreted as the nurses being vaccinated.

- Thank you for the suggestion, the sentence has been revised to clarify.

The second statement in the introduction is unclear, "In Europe alone, 14 600 cases were reported in 2017 and in the WHO European Region more than 82 000 cases were reported [1, 2]". Is there an explanation for the difference in these two figures - are they both needed?

- Thank you for your suggestion. The WHO European region includes more countries. We have revised the sentence as suggested and removed one of the figures.

Finally, the introduction is not well-structured, which makes it difficult to follow. Please see the attachment for a suggestion as to how this could be done more clearly.

- Thank you for reading the introduction carefully. We have read your suggestion thoroughly and revised as suggested for the majority of the changes made.